# Decreased ENSO post-2100 in response to formation of a permanent El Niño-like state under greenhouse warming

Tao Geng [1], Wenju Cai [1,2,3,4] ✉, Fan Jia [5] ✉ & Lixin Wu [1,2]

Under transient greenhouse warming, El Niño-Southern Oscillation (ENSO) is projected to increase pre-2100, accompanied by an easier establishment of atmospheric convection in the equatorial eastern Pacific, where sea surface temperature (SST) warms faster than surrounding regions. After 2100, how ENSO variability may change remains unknown. Here we find that under a high emission scenario, ENSO variability post-2100 reverses from the initial increase to an amplitude far smaller than that of the 20th century. The fast eastern warming persists and shrinks the equatorial Pacific non-convective area, such that establishing convection in the non-convective area, as during an El Niño, requires smaller convective anomaly, inducing weaker wind anomalies leading to reduced ENSO SST variability. The nonlinear ENSO response is thus a symptom of the persistent El Niño-like warming pattern. Therefore, the oscillatory ENSO impact could be replaced by that from the permanent El Niño-like mean condition with cumulative influences on affected regions.

El Niño-Southern Oscillation (ENSO) is the most consequential mode of climate variability[1–3]. During El Niño, warm anomalies in the central and eastern equatorial Pacific weakens the west-minus-east zonal SST gradient along the equator[1,4]. The associated decrease in the trade winds weakens upwelling and deepens the thermocline in the equatorial eastern Pacific, in turn intensifying the warm anomaly, in a positive feedback process referred to as Bjerknes feedback[5]. During an extreme El Niño event, as in 1997/98, substantial warming in the equatorial central and eastern Pacific erases much of the climatological west-minus-east and meridional off-equator-minus-equator SST gradients along the equatorial Pacific, such that western Pacific and the Inter-Tropical Convergence Zones (ITCZ)[6,7], and South Pacific Convergence Zone (SPCZ)[8,9] move toward the equatorial eastern Pacific, where atmospheric convection establishes[6,8,10,11]. The climatological gradients of SSTs thus indicate the potential intensity of El Niño events and the extent of their reorganization of atmospheric convection.

Under a transient greenhouse warming, majority of climate models project an increase in ENSO SST variability in the 21st century from that in the 20th century[12–16], accompanied by persistent unidirectional mean state changes[14]. The equatorial eastern Pacific warms faster than the surrounding regions[14,17–19], increasing the ease at which atmospheric convection establishes in the equatorial eastern Pacific[6,10], and swings of the ITCZ and the SPCZ toward the equator[8,11,20]. The upper ocean warms faster than the ocean below[17], enhancing oceanic stratification, which intensifies ocean-atmosphere coupling[13,21–23]. In addition, warming background SSTs strengthen tropical-extratropical two-way interactions[24,25]. Prior to 2100, the conducive mean state changes tend to lead to an increase in intensity and frequency of strong ENSO events[12–15,21], although the change differs vastly across models[26,27], which is partially attributed to internal variability[26–28]. After 2100, how ENSO SST variability may change remains unknown.

In a small subset of models of the Coupled Model Intercomparison Project Phase 5 (CMIP5) (ref. 29), ENSO variability over the

[1]Laoshan Laboratory, Qingdao, China. [2]Frontiers Science Center for Deep Ocean Multispheres and Earth System (FDOMES) and Key Laboratory of Physical Oceanography, Ocean University of China, Qingdao, China. [3]State Key Laboratory of Marine Environmental Science & College of Ocean and Earth Sciences, Xiamen University, Xiamen, China. [4]State Key Laboratory of Loess and Quaternary Geology, Institute of Earth Environment, Chinese Academy of Sciences, Xi'an, China. [5]Key Laboratory of Ocean Observation and Forecasting and Key Laboratory of Ocean Circulation and Waves, Chinese Academy of Sciences, and Laoshan Laboratory, Qingdao, China. ✉e-mail: wenju.cai@csiro.au; jiafan@qdio.ac.cn

20th to the 21st century increases initially but commences a reversal around 2040 (ref. 30). The reversal is due to a warming differential in the equatorial Indo-Pacific from a faster warming in the equatorial Pacific to a faster warming in the Indian Ocean. Models under an instantaneous doubling or quadrupling of $CO_2$ generate a reduction in ENSO variability in the quasi-stable state[31], directly proportional to the amount of $CO_2$ increase, such that ENSO almost vanishes under a $CO_2$ concentration four times the preindustrial level[32]. Due to limitations such as the short time length of simulation[30] and idealized implementation of $CO_2$ forcing[31,32], how ENSO variability under a persistent greenhouse warming may evolve into the future beyond 2100 is unclear.

Here, using 17 available models forced under a high emission scenario, in which atmospheric $CO_2$ increases to more than seven times the pre-industrial level by 2250 (ref. 33), we examine the evolution of ENSO into 2300. We find that majority of the models simulate a reversal of ENSO variability, from an initial increase to a subdued amplitude far smaller than that of the 20th century, amid development of an El Niño-like background warming pattern, previously defined as a substantial weakening in climatological zonal and meridional SST gradients[34–36] and resembling a permanent El Niño-like mean condition, suggested to have occurred during the earlier Pliocene warm period[34,36–38].

## Results

### Reversal under unidirectional mean state changes
Although a large number of CMIP5 and CMIP6 models are forced under various emission scenarios, only a limited number of models extend beyond 2100 (refs. 29,39). The number is largest under a high emission scenario, with 9 CMIP5 models under Representative Concentration Pathway 8.5 (RCP8.5) and 8 CMIP6 models under Shared Socioeconomic Pathway 585 (SSP585) (see "Data and processing" in Methods). Under these high emission scenarios, $CO_2$ increases to ~2200 ppm by 2250 from a constant pre-industrial level of ~285 ppm[33]. We use all the available models and focus on December-January-February (DJF) in which ENSO typically matures. To test the sensitivity of our results to emission scenarios, we also analyze 14 available models under a strong mitigation emission scenario (RCP26 for CMIP5 and SSP126 for CMIP6; Supplementary Table 1), in which $CO_2$ ramps down from ~470 ppm in 2050 to ~400 ppm by 2250 (ref. 33).

For each model, DJF SST anomalies in the equatorial central and eastern Pacific (the 'Niño3.4' region; 5ºS-5ºN, 170ºW-120ºW) are first constructed with reference to a 51-year running mean DJF climatology from 1860 to 2299, a period common to all models. To eliminate any influence from variability on decadal and longer time scales, we further subtract an 11-year running mean of the DJF anomaly from the original anomaly time series. Thus, the resultant anomaly time series only contains interannual variability. Evolution of ENSO variability, defined as standard deviation of the DJF Niño3.4 SST anomaly, is calculated first in each model and then averaged across the models, in a 51-year running window from 1860 to 2299. Our results are not sensitive to the length of running windows or way of detrending to compute the anomalies (see "Data and processing" in Methods). We also apply this method to the multi-century-long pre-industrial experiment (piControl) in each model to gauge natural variations of ENSO.

Projected changes in the Pacific mean state, expressed in terms of 51-year running means, persist unidirectionally into the end of 23rd century, including the faster warming in the equatorial eastern Pacific, faster warming in the equatorial than the off-equatorial, weakening of the trade winds, and intensification of the upper ocean stratification[17,18]. In association, there is a weakening in the Walker circulation and a shallowing of the equatorial Pacific thermocline (Fig. 1a, b). Despite the unidirectional background changes into the 23rd century, ENSO variability undergoes an initial increase followed by a subsequent reversal (Fig. 1c, black). Such a reversal is seen in the

majority of models available, although the timing of reversal differs (Supplementary Fig. 1).

To understand the contribution to the evolution of ENSO variability by El Niño frequency and El Niño amplitude, we calculate 51-year running averages of El Niño frequency defined as when the DJF Niño3.4 index exceeds a value of 0.75 standard deviation (s.d.)[40,41] calculated from the corresponding running periods, and running averages of amplitude of the El Niño events. The reversal is not due to the frequency (Fig. 1c, brown), but to the averaged intensity of the events, particularly, strong El Niño events defined as when the Niño3.4 index exceeds a value of 1.50 s.d. (Fig. 1d).

As a sensitivity test, we use simulation of nonlinear Bjerknes feedback to select models for an assessment. As a consequence of the feedback, warm anomalies and weakened equatorial Pacific SST gradients of an El Niño establish atmospheric convection in the usually cold and dry equatorial eastern Pacific, leading to a large increase in rainfall, therefore a positive skewness in rainfall. Using a skewness value greater than 1.0 and model ability to simulate extreme El Niño as in a previous study[6], we select 9 models (models marked by a *, Supplementary Fig. 2). The result based on the 9 selected models reinforces the finding above, in particular, the initial increase in ENSO SST variability is stronger than that based on all models (Supplementary Fig. 3). The initial increase is consistent with the mean state changes but the reversal is not, highlighting a nonlinear response of ENSO to a persistent greenhouse warming, as we describe below.

### A nonlinear response to global warming
A cause for the initial ENSO enhancement is the fast warming in the equatorial eastern Pacific, which increases the ease at which stronger atmospheric convection is established in the equatorial eastern Pacific[6,42]. There has been a debate as to how the background mean SST of the tropical Pacific may respond to greenhouse warming[43,44]. Though different from the recently observed[43–45], evidence suggests that the enhanced eastern Pacific warming pattern in the future is possible as greenhouse effects progressively overwhelm other factors, such as decadal climate variability, which in observations could temporarily mask the warming signal[46,47]. The fast eastern Pacific warming also contributes to an overall stronger equatorial warming than the off-equatorial region, and increases the proximity of the convergence zones toward the equator, facilitating their movement to the equator. These changes induce wind anomalies that subsequently reinforce the warm anomalies through the Bjerknes feedback.

We describe the proximity of off-equatorial convergences zone to the equator using an index that combines the latitudinal position of the ITCZ and the SPCZ (see "Proximity of the convergence zones to the equator" in Methods). To enhance inter-model comparability, both the proximity and ENSO variability are appropriately scaled by their respective piControl level expressed in percentage.

For each model, time series of 51-year running averages of the latitude of the ITCZ and the SPCZ centers are constructed before time series of a multi-model average is constructed (Fig. 2a, b). The two convergence zones move toward the equator; consistently, the proximity to the equator of the two convergence zones combined increases throughout the 400 years, expressed as a downward trend (Fig. 2b). A close proximity is initially conducive to an increase in ENSO variability as seen in a relationship between ENSO SST variability and the proximity calculated over periods of a century.

Based on samples from the piControl and the 20th century, during which time the proximity increases mildly, the relationship shows that a closer proximity favors ENSO variability (Fig. 2c). This relationship is systematic, with a correlation of −0.50, statistically significant above the 95% confidence level. However, based on samples over the 21st, the 22nd and the 23rd century, during which time the proximity increases substantially, a closer proximity is instead associated with a greater reduction in ENSO variability, with a statistically significant

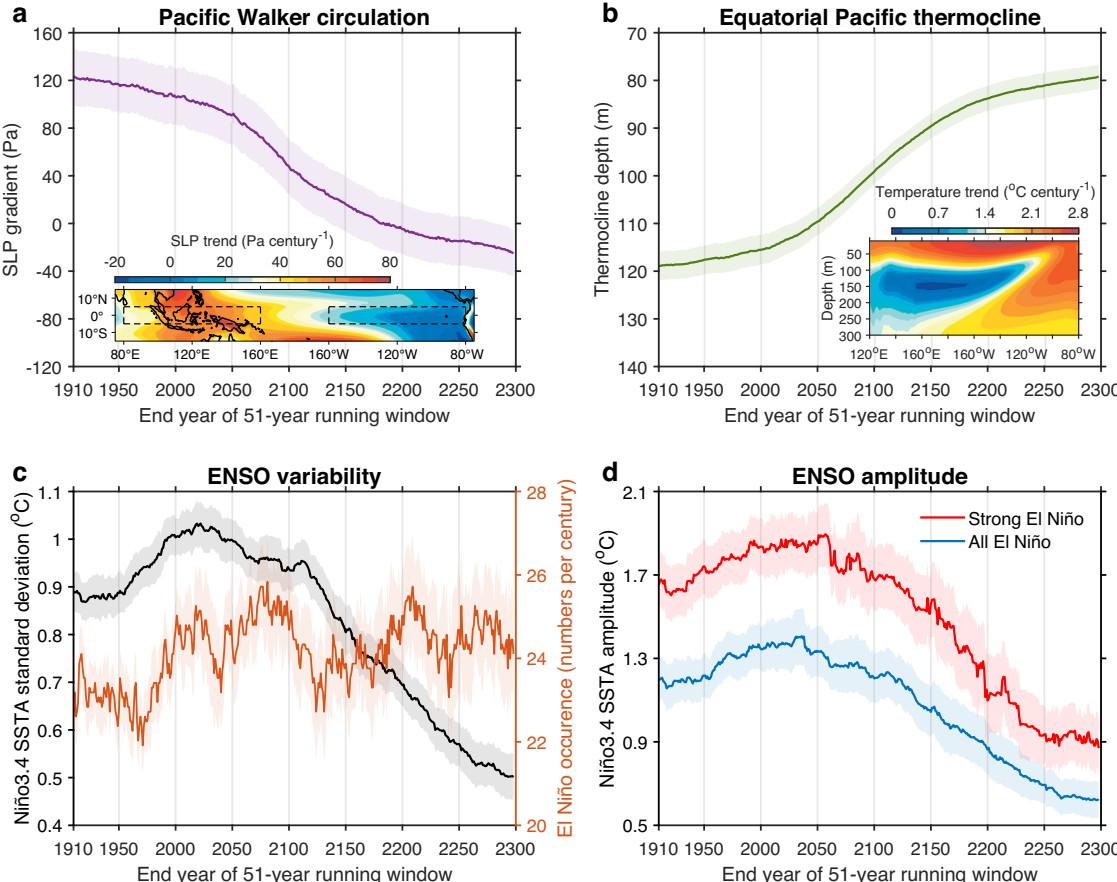

**Fig. 1 | Evolution of Pacific mean state and El Niño-Southern Oscillation (ENSO) variability under persistent greenhouse warming. a** 51-year running mean intensity of Pacific Walker circulation[59], calculated as sea level pressure (SLP) gradient between central-eastern (5ºS-5ºN, 160ºW-80ºW) and Indo-western Pacific (5ºS-5ºN, 80ºE-160ºE), recorded as the end year of the windows, from 1860 to 2299 under high-emission scenarios. Also shown is a multi-model mean spatial pattern of SLP linear trends over the period (1860–2300), with dashed boxes indicating the regions used to calculate the SLP gradient. **b** As in **a**, but for thermocline depth in the equatorial Pacific (5ºS-5ºN, 120ºE-80ºW). The vertical temperature trend pattern is averaged over 5ºS-5ºN. **c** 51-year running standard deviation of Niño3.4 sea surface temperature (SST) anomaly (black) and number of El Niño events (brown). **d** 51-year running mean amplitude of Niño3.4 SST anomaly for strong (red) and all (blue) El Niño events. All the indices are calculated over the December-February (DJF) season. Solid lines and shadings indicate multi-model mean and 1.0 standard deviation of a total of 10,000 inter-realizations based on a Bootstrap method, respectively. Despite unidirectional background mean state changes, ENSO variability undergoes an initial increase followed by a subsequent reversal, dominated by amplitude of El Niño rather than El Niño frequency.

correlation of 0.67 (Fig. 2d). The contrasting ENSO responses are more evident in the 9 selected models (Supplementary Fig. 3). Thus, ENSO variability changes nonlinearly with the proximity of convergence zones to the equator.

### Decreased potential intensity and non-convective area

In addition to driving the increased proximity, the fast warming in the equatorial eastern Pacific induces a reduction in the climatological zonal and meridional SST gradients, hence in the potential El Niño intensity. In observation, strong El Niño warm anomalies are usually associated with a substantial decrease in the equatorial west-minus-east, south-minus-east and north-minus-east SST gradients, which determine the extent to which equatorial western Pacific convection moves to the east and the SPCZ and the ITCZ shift toward the equator, respectively (Supplementary Fig. 4). We therefore construct the potential El Niño intensity taking into account of both the zonal and meridional SST gradients (see "Potential El Niño intensity" in Methods). The same fast warming in the east also shrinks the non-convective area. We calculate non-convective area in the equatorial Pacific as the areal coverage where SSTs are lower than the contemporary tropical 20ºS-20ºN mean[48], expressed in terms of percentage of the equatorial area (5ºS-5ºN, 120ºE-80ºW).

Evolution in terms of 51-year running averages of both the potential intensity and the non-convective area (Fig. 3a, b) and in terms of centennial averages (Supplementary Fig. 5), shows a slight decrease during the pre-2100 period followed by a substantial reduction during the 22nd and 23rd centuries. Their coherent changes are seen in the multi-model average and in individual models (Supplementary Fig. 6). The shrinking in non-convective area means that the establishment of atmosphere convection in the non-convective area, as during an El Niño, requires increasingly smaller SST or convective anomalies, consistent with the reduced potential intensity. The reduction in the potential intensity and non-convective area thus gives rise to a curtailing effect through reduced wind anomalies.

The curtailing effect on El Niño amplitude is initially weak with no systematic relationships before 2100 (Supplementary Fig. 7). Subsequently, it competes with the conducive effects including an enhanced air-sea coupling[13,21], tropical-extratropical interactions[24,25], and the easier establishment of convection in the east. Eventually, the curtailing effect dominates, leading to the reduction in ENSO variability, despite the increasing proximity, which continues to favor frequent El Niño events (Fig. 1c, brown). Consequently, in models with a smaller potential intensity in the 22nd or the 23rd century, maximum ENSO amplitude is systematically weaker, with an inter-model correlation of 0.76 (Fig. 3c). Similarly, in models with a greater shrinking of the non-

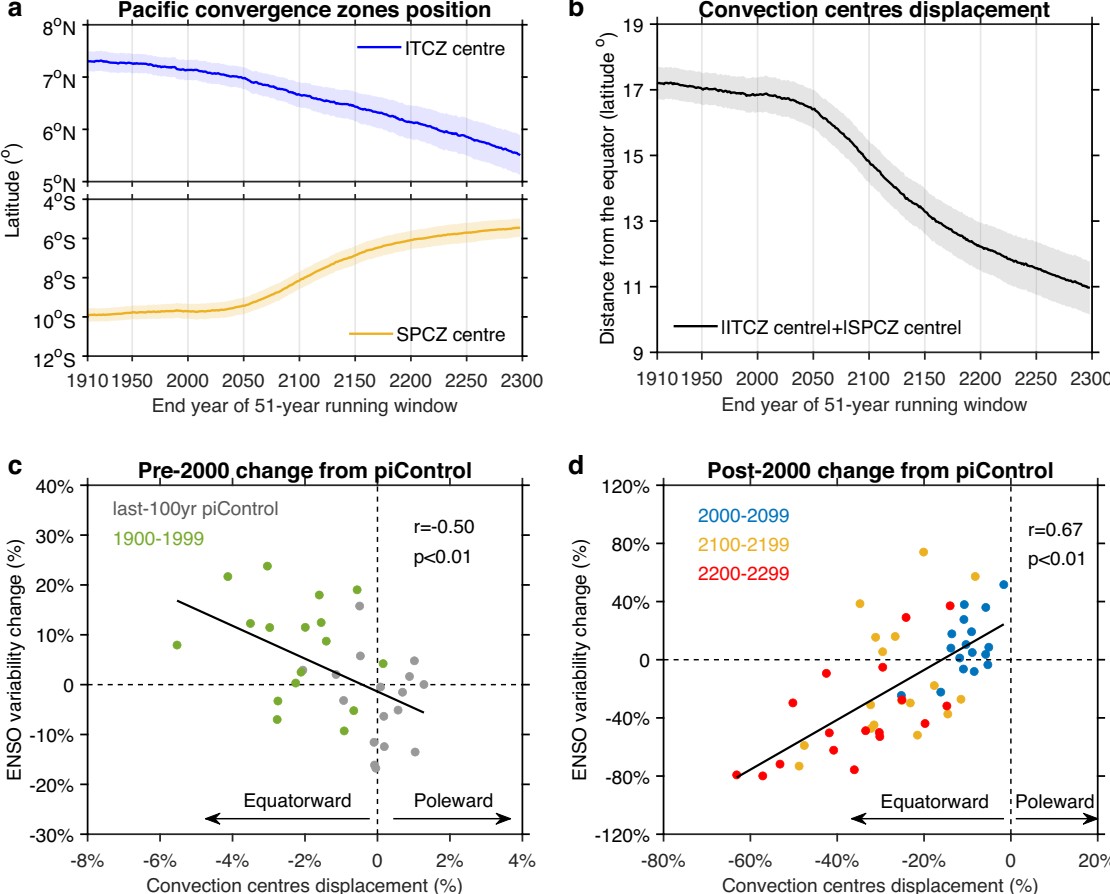

**Fig. 2 | Nonlinear El Niño-Southern Oscillation (ENSO) response to equatorward movement of Pacific convergence zones. a** 51-year running mean latitudinal position of the Inter-Tropical Convergence Zone (ITCZ; blue), and the South Pacific Convergence Zone (SPCZ; yellow) centers in October-February (ONDJF) from 1860 to 2099 under high-emission scenarios. Years on the x-axis denote the end year of the running window. Solid lines and shadings indicate multi-model mean and 1.0 standard deviation of a total of 10,000 inter-realizations based on a Bootstrap method, respectively. **b** As in **a**, but for the proximity of the convergence zones to the equator, calculated as the sum of latitude of the ITCZ and the SPCZ centers. **c** Inter-model relationship between the mean ONDJF latitudinal position of ITCZ

and SPCZ with December-February (DJF) Niño3.4 sea surface temperature (SST) standard deviation, both referenced to a piControl 100-year rolling mean and expressed in percentage, in the last 100 years of piControl (gray dots) and the 20th century (green dots). Negative values of the convection centers displacement indicate an equatorward movement of the Pacific convergence zones. A linear fit (solid black line) is displayed together with correlation coefficient *r* and corresponding *p* value. **d** Same as **c**, but for the 21st (blue dots), 22nd (yellow dots) and 23rd (red dots) centuries. ENSO variability changes nonlinearly with the proximity of Pacific convergence zones to the equator.

convective area, El Niño amplitude is systematically weaker, with an inter-model correlation of 0.74 (Fig. 3d). The controlling effect of the decreased potential intensity and reduced non-convective area, which becomes established after 2100, is exerted by limiting El Niño convective and wind anomaly and inhibiting thermocline-wind coupling, as illustrated below.

### Weakened ENSO a symptom of incipient permanent El Niño-like mean condition

The shrinking of the non-convective area eventually curtails amplitude of El Niño convective anomalies. In models with a greater shrinking, amplitude of El Niño convective anomalies, measured by outgoing longwave radiation at the top of the atmosphere, is systematically smaller (Fig. 4a). The weakened convective anomalies in turn lead to a reduction in wind anomalies over the equatorial central and eastern Pacific (Fig. 4b). The smaller wind anomalies feed into the Bjerknes feedback, driving a weaker response of the thermocline to the winds[30,49,50], ultimately leading to a weaker El Niño.

The dynamical coupling between thermocline and wind, measured by the regression of zonal thermocline slope anomalies onto wind anomalies[30] (see "Thermocline-wind coupling" in Methods),

undergoes a similar pre-2100 increase followed by a substantial post-2100 reduction (Fig. 4c). The post-2100 reduction is statistically significant above the 95% confidence level (Fig. 4d), according to a Bootstrap method (see "Statistical significance test" in Methods). Models that simulate a smaller wind anomaly during the 22nd and 23rd centuries systematically generate a weaker thermocline total response to wind (Fig. 4e), which eventually leads to a weaker El Niño SST variability through the thermocline feedback[30,49,50] (Fig. 4f). A weaker El Niño in turn generates a smaller upper ocean heat discharge of the equatorial Pacific[51], and hence a correspondingly weaker La Niña. The subdued ENSO amplitude is most conspicuous in two models (CESM2-WACCM and GISS-E2-R, Supplementary Table 1), in which the potential intensity and ENSO variability in the 23rd century decreases dramatically to 11.32% and 17.67% of the 20th century, respectively, for one model (Fig. 5a).

ENSO rectification, in which higher ENSO variability rectifies on the mean state via nonlinear oceanic temperature advection leading to a larger eastern Pacific warming[52], would not explain the systematic relationship between decreased ENSO variability and enhanced eastern Pacific warming after 2100 (Figs. 3c and 5a). The reduction in ENSO variability is thus a symptom of a developing permanent El Niño-like

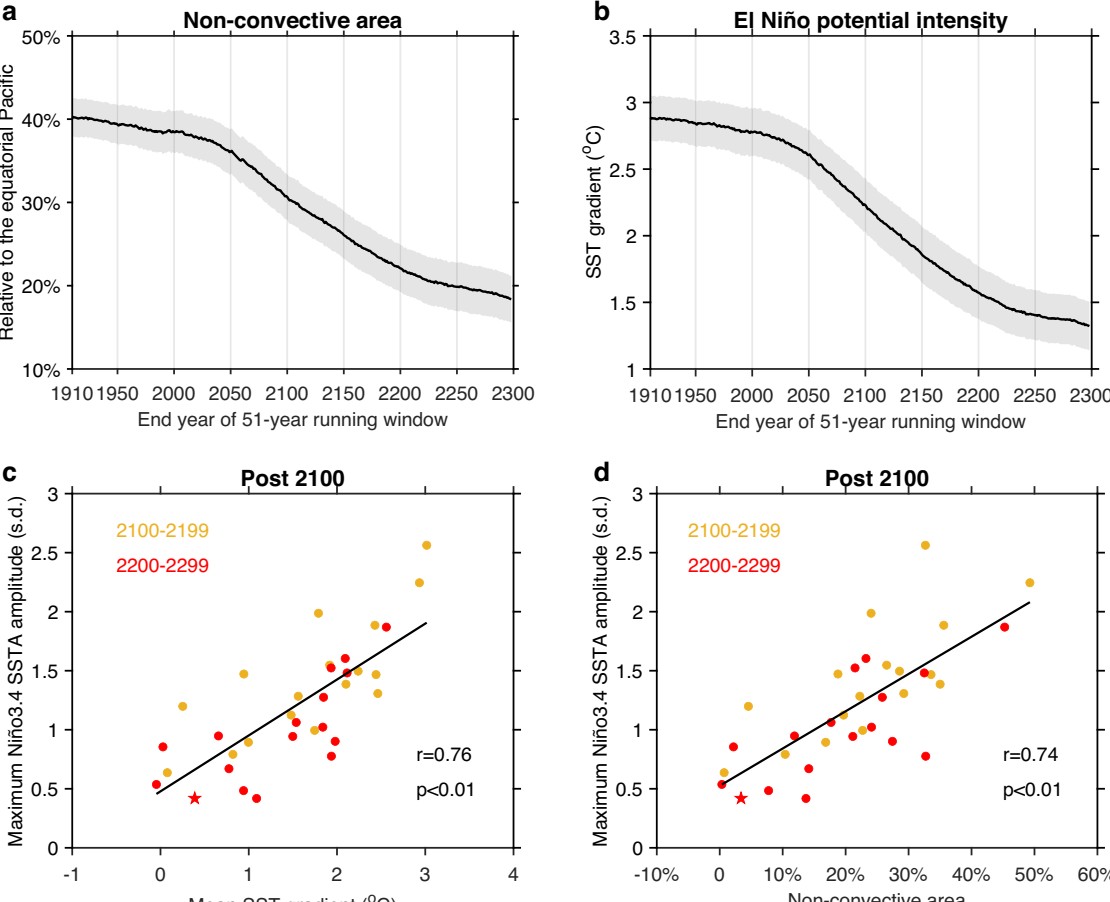

**Fig. 3 | Decreased non-convective area constrains El Niño intensity after 2100. a** 51-year running mean area of non-convective region as percentage of the equatorial Pacific areal coverage (5°S-5°N, 120°E-80°W), calculated as the areal coverage where sea surface temperatures (SSTs) are lower than the tropical (20°N-20°S) average, from 1860 to 2300 under high-emission scenarios. Years on the x-axis denote the end year of the running window. Solid lines and shadings indicate multi-model mean and 1.0 standard deviation of a total of 10,000 inter-realizations based on a Bootstrap method, respectively. **b** As in **a**, but for mean SST gradient that sets the potential intensity of El Niño, calculated as the SST difference between an average over west (2.5°S-2.5°N, 120°E-180°), north (5°N-10°N, 150°W-90°W) and south (5°S-10°S, 160°E-140°W) equatorial Pacific and that over the east (2.5°S-2.5°N,

150°W-90°W) equatorial Pacific. **c** Inter-model relationship between the mean SST gradient and maximum El Niño amplitude averaged over the top 10 strongest Niño3.4 SST anomaly in the 22nd (yellow dots) and the 23rd (red dots) centuries. To enhance inter-model comparability, the Niño3.4 SST is normalized with its 400-year (1900–2299) standard deviation (s.d.). Linear fit (solid black line) is displayed together with correlation coefficient $r$ and corresponding $p$ value. **d** Same as **c**, but for non-convective area and maximum El Niño amplitude. All the indices are calculated over the DJF season. Red star denotes the model CESM2-WACCM. Post-2100 El Niño-Southern Oscillation (ENSO) amplitude is curtailed by shrinking non-convective area and decreasing El Niño potential intensity.

mean condition characterized by the collapsing west-minus-east and meridional SST gradients (red and blue curve, Fig. 5a). The "El Niño-like" mean condition should not be taken as a state where all climate anomalies resemble those of an El Niño, or that the associated mechanisms for the changes are the same as those of El Niño[53]. Instead, an "El Niño-like" mean condition here is characterized by a substantially weakened west-minus-east and meridional SST gradients, with only some anomalies resembling those of an El Niño. Such a permanent El Niño-like mean condition was speculated to have occurred during the earlier Pliocene period ~4.5 to 3.0 million years ago[34,36–38].

The weakened ENSO toward a collapse could be mistaken as good news given the extreme impacts of strong El Niño and strong La Niña events. However, this is a situation in which ENSO is replaced by a permanent El Niño-like condition with its own adverse impacts, in many respects akin to those during an El Niño. To gauge this impact, for each model, we construct composite rainfall anomalies referenced to its respective historical experiment in the 20th century, and generate an average over five models simulating top-five largest reductions in potential intensity, that is, with a well-developed permanent El

Niño-like mean condition, and another average over five models producing bottom-five smallest reductions, that is, with a least-developed permanent El Niño-like mean condition. In the well-developed group, La Niña rainfall anomalies referenced to the 20th century climatology show, instead, El Niño-like anomaly characteristics (Fig. 5b, c). For example, the wet anomalies over northern and northeastern Australia during a historical La Niña event almost disappear, and the dipole pattern of northern-dry and southern-wet anomalies over south Africa, northern-wet and southern-dry anomalies over eastern South America, or the equatorial-dry and off equatorial-wet anomalies, simulated during the 20th century La Niña, reverses, dominated by impact from the permanent El Niño-like pattern (Supplementary Fig. 8a). The El Niño-like impacts during La Niña years are also seen in the least-developed group, though relatively weaker (Supplementary Fig. 8b–d). Thus, a consequence of a permanent El Niño-like mean state is that even during La Niña years, many ENSO-affected regions would experience an El Niño-like impact seen in the historical period.

The prospect of a permanent El Niño-like mean condition disappears under the SSP126 emission scenario. We examine all available 14 models forced under the scenario that extend to 2300. There is not

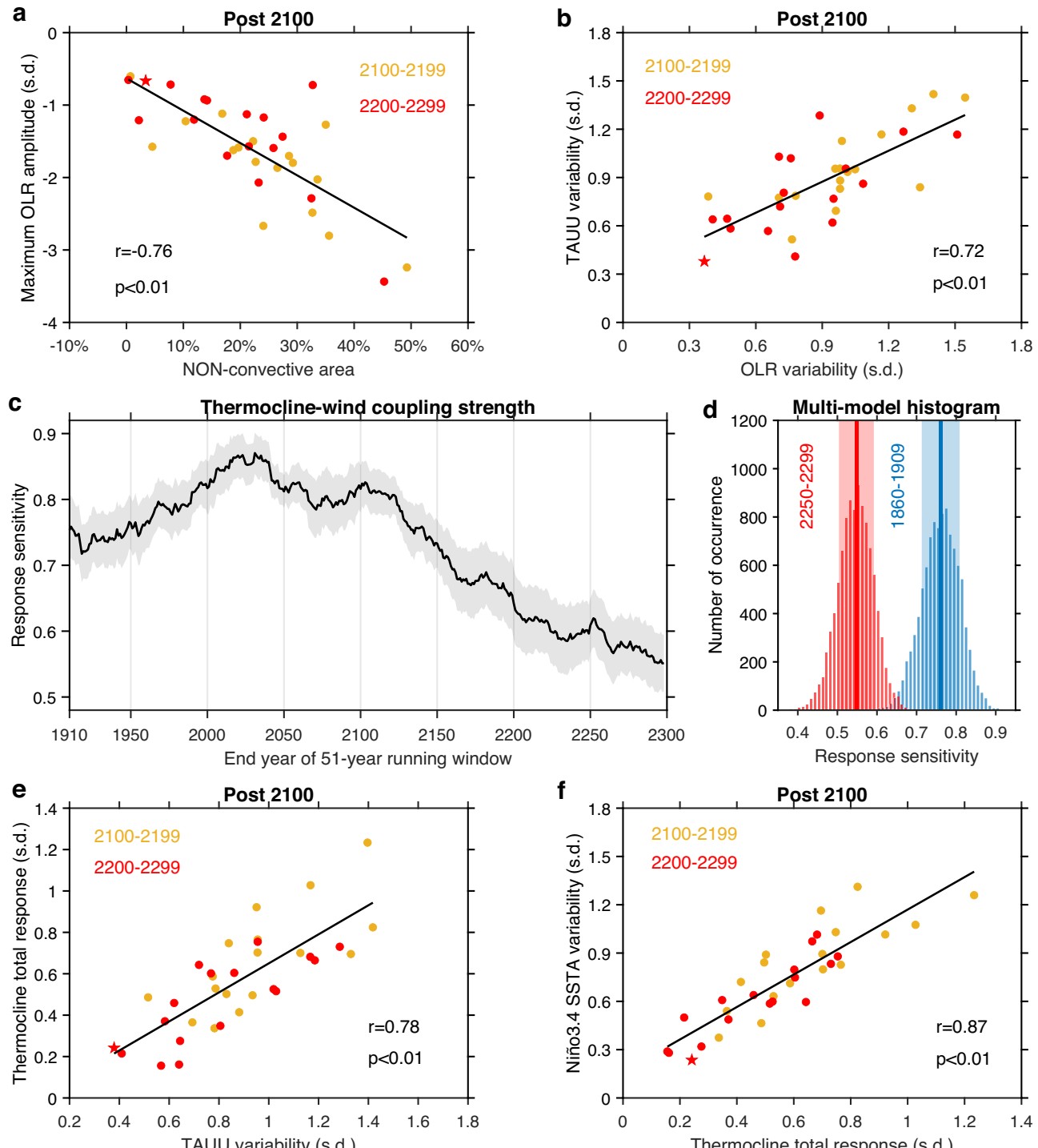

**Fig. 4 | Reduced thermocline-wind coupling curtails post-2100 El Niño-Southern Oscillation (ENSO) variability. a** Inter-model relationship between non-convective area and maximum convection amplitude averaged over the top 10 strongest outgoing longwave ration (OLR) anomaly in the central equatorial Pacific (5°S-5°N, 170°E-130°W) in the 22nd (yellow dots) and the 23rd (red dots) centuries. The OLR index is normalized by its whole 400-year (1900–2299) standard deviation (s.d.). Linear fit (solid black line) is displayed together with correlation coefficient *r* and corresponding *p* value. **b** As in **a**, but for standard deviations (s.d.) of OLR and zonal wind stress (TAUU) anomalies in the central equatorial Pacific. All the indices are calculated over the December-February (DJF) season. Red star denotes the model CESM2-WACCM. **c** 51-year running mean response sensitivity coefficient from regression of normalized zonal thermocline slope (east minus west) anomalies onto normalized TAUU anomalies, from 1860 to 2300 under high-emission scenarios. Years on the x-axis denote the end year of the running window. **d** Multi-model histograms of 10,000 realizations of a Bootstrap method for the thermocline slope response sensitivity to wind in 1860–1909 (blue bars) and 2250–2299 (red bars). Solid lines and shadings in **c**, **d** indicate multi-model mean and 1.0 standard deviation of a total of 10,000 inter-realizations based on a Bootstrap method, respectively. **e**, **f** Same as **b**, respectively, but for **e** inter-model relationship between TAUU variability and total response of thermocline slope to wind, calculated as the thermocline response sensitivity coefficient multiplied by the standard deviation of TAUU, and **f** inter-model relationship between total response of thermocline slope to wind and normalized Niño3.4 sea surface temperature (SST) standard deviation. These results indicate that reduced wind variability from the decreased potential intensity and non-convective area leads to reduced post-2100 ENSO variability through weakened thermocline response to the wind, feeding into dynamical ocean-atmosphere coupling.

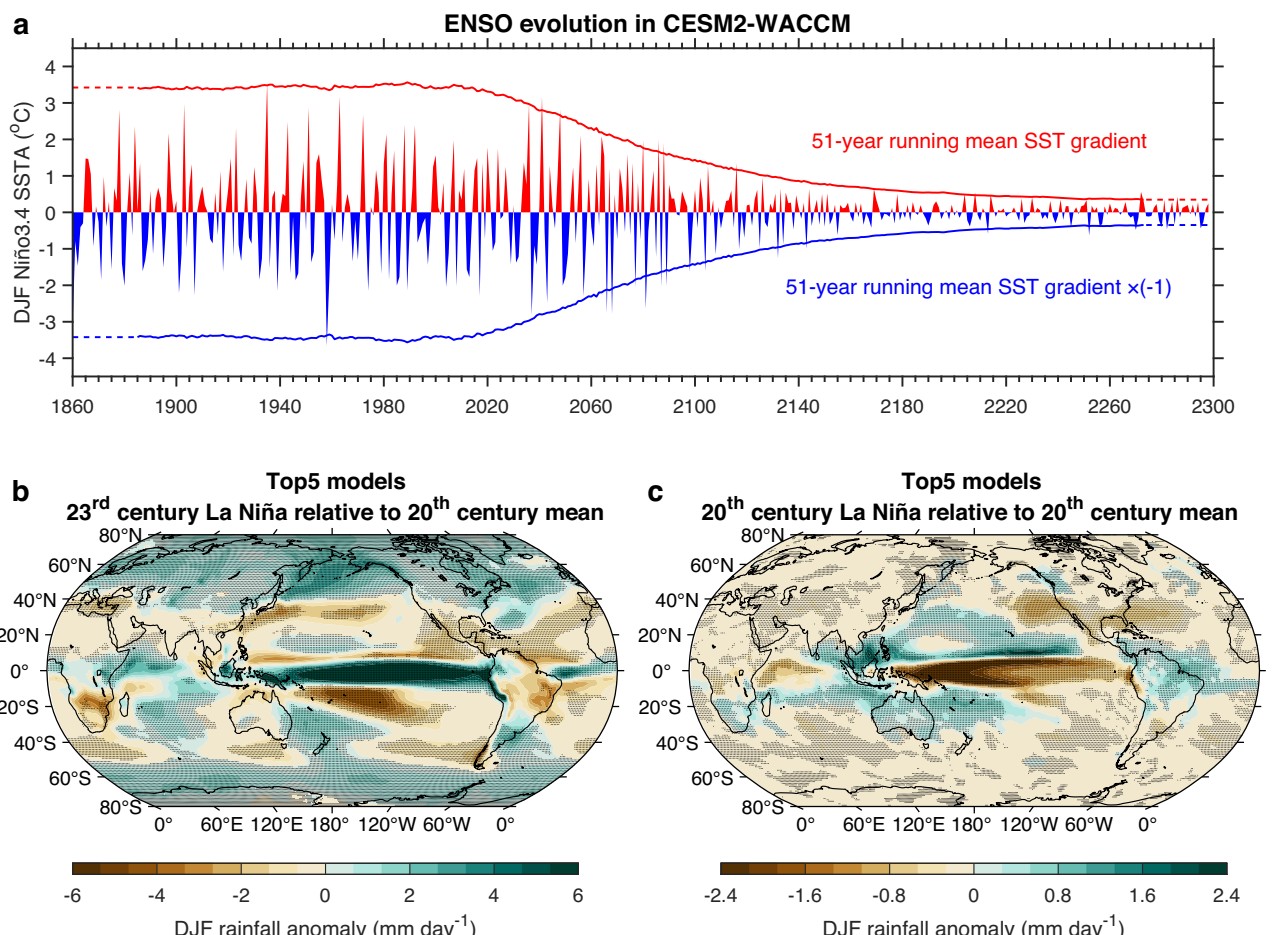

**Fig. 5 | Weakened post-2100 El Niño-Southern Oscillation (ENSO) amid a transition into a permanent El Niño-like mean condition. a** Time series of December-February (DJF) Niño3.4 sea surface temperature (SST) anomaly (bar) and 51-year running mean SST gradient (lines) over DJF in the model CESM2-WACCM. The running SST gradient is plotted against the central (26th) year of the 51-year sliding window, with the first and the last 25 years shown in dashed lines for illustration. **b** Multi-model mean DJF rainfall anomalies during La Niña years in the 23rd century (2200–2299) relative to the 20th century (1900–1999) DJF climatology in the top 5 models simulating the largest reductions in potential El Niño intensity. **c** As in **b**, but for 20th century La Niña rainfall anomalies. The dotted area indicates regions where the rainfall anomaly is statistically significant above the 95% confidence level according to a Bootstrap test. Despite a reduction in variability, ENSO-induced global impacts could be replaced by a permanent El Niño-like mean condition after 2100.

a persistent decrease in ENSO variability toward 2300; the reduction in the potential El Niño intensity or the climatological non-convective area, stops by 2090, and stabilises thereafter, in a sharp contrast to the evolution under SSP585 (Supplementary Fig. 9).

## Discussion

Our finding of a reversal in ENSO evolution from an initial increase to a subdued or collapsed amplitude toward 2300 highlights a strong nonlinear response of ENSO to persistent greenhouse warming. Under transient greenhouse warming, the fast warming in the equatorial eastern Pacific, though initially contributing to an increase in ENSO variability, sows the seeds of a subsequent ENSO decrease by reducing the potential intensity and shrinking the climatological non-convective area of the equatorial Pacific. Establishing atmospheric convection in the reduced non-convective area induces small convective and wind anomalies that feed into the Bjerknes feedback. The associated curtailing effect eventually leads to the ENSO reduction. Although ENSO from a reduction to an eventual collapse is seen in some models, the likelihood in other models requires experiments integrated beyond the 23rd century, but such an evolution represents no good news, because it means that the oscillatory nature of ENSO impacts would be replaced by a quasi-permanent condition that is similar to an El Niño in many respects, therefore with a cumulative impact on affected regions.

## Methods

### Data and processing

We use SST reanalysis data from ERSST v5 (Extended Reconstructed Sea Surface Temperature version 5)[54] and satellite-based rainfall data from GPCPv2.3 (Global Precipitation Climatology Project)[55], for the period from 1979 to 2023. We focus on December-January-February (DJF) in which ENSO typically matures. DJF anomalies of SST and rainfall is constructed with reference to the climatological DJF mean of 1979–2023 and quadratically detrended.

To examine ENSO response to persistent greenhouse warming beyond the 21st century, we take nine CMIP5 models[15] and eight CMIP6 models[39] that are available so far (Supplementary Table 1), forced under historical anthropogenic and natural forcings to 2005 for CMIP5 and 2014 for CMIP6, respectively, and thereafter future greenhouse gas forcing till 2300 under the Representative Concentration Pathway 8.5 (RCP85) for CMIP5 and equivalent Shared Socioeconomic Pathway 5-8.5 (SSP585) emission scenario for CMIP6. We focus on the 1860–2299 period that is common to all models. Monthly outputs of surface temperature, SST, zonal wind stress, ocean temperature, sea level pressure, outgoing longwave radiation and rainfall are utilized, to calculate global mean temperature, ENSO variability, zonal wind variability, equatorial thermocline depth defined as the depth of maximum vertical temperature gradient, Walker circulation intensity,

atmosphere convection variability and location of Pacific convergence zones, respectively. Prior to analysis, outputs of each model are re-gridded into a common 1º × 1º resolution.

For each model, DJF anomalies are first constructed with reference to a 51-year running mean DJF climatology from 1860 to 2299. The choice of using 51-year base period to calculate anomaly is to accord with the 51-year running window used to examine ENSO variability change. To eliminate any influence from variability on decadal and longer time scales, we further subtract an 11-year running mean of the DJF anomaly from the original anomaly time series. Thus, the resultant anomaly time series only contains interannual variability. We apply this method to all variables and in the pre-industrial experiment (piControl) with a multi-century-long simulation. Change in ENSO variability, calculated as DJF SST standard deviation in the Niño3.4 region (5ºS-5ºN, 170ºW-120ºW), is calculated with reference to the corresponding piControl mean level in each model and expressed in percentage.

We have tested that using different ways to calculate anomaly, such as a polynomial fit to detrend[56] or NOAA's method to compute Oceanic Niño Index (https://origin.cpc.ncep.noaa.gov/products/analysis_monitoring/ensostuff/ONI_v5.php), or using different length (11-yr, 31-yr and 71-yr) of running windows, does not qualitatively alter our results (Supplementary Figs 10 and 11). To test the sensitivity of our results to emission scenarios, 14 available models under a strong mitigation emission scenario (RCP26 for CMIP5 and SSP126 for CMIP6; Supplementary Table 1) are also analyzed. Other warming scenarios are not used due to the scarcity of data because few extend beyond 2100.

### Proximity of the convergence zones to the equator

We measure the proximity of Pacific convergence zones to the equator by summing the absolute latitudinal position of ITCZ and SPCZ centers (|ITCZ centre|+|SPCZ centre|). The ITCZ position (|ITCZ centre|) is defined as the average latitude over which rainfall in the tropical north Pacific Ocean (0-20ºN, 120ºE-90ºW) is greater than 80% of the maximum zonal averaged rainfall[57]. Similarly, the SPCZ position (|STCZ centre|) is defined as the average latitude over which rainfall in the tropical south Pacific Ocean (0-30ºS, 120ºE-90ºW) is greater than 80% of the maximum zonal averaged rainfall. By definition, the proximity index is positive and a decrease (increase) in the index means an equatorward (poleward) movement of the convergence zones.

### Potential El Niño intensity

During an extreme El Niño event, as seen in 1997/98, substantial SST warming in the equatorial central and eastern Pacific erases much of the climatological west-minus-east and meridional off-equator-minus-equator SST gradients along the equatorial Pacific, such that western Pacific and the Inter-Tropical Convergence Zones (ITCZ)[6,7], and South Pacific Convergence Zone (SPCZ)[8,9] migrate toward the equatorial eastern Pacific, where the atmospheric convection establishes[6,8,10]. The climatological gradients of SSTs thus indicate a potential SST intensity of El Niño events and the extent of their reorganization of atmospheric convection. We define the potential El Niño intensity as an SST gradient index, calculated as the SST difference between an average over the west (2.5ºS-2.5ºN, 120ºE-180º), north (5ºN-10ºN, 150ºW-90ºW) and south (5ºS-10ºS, 160ºE-140ºW) equatorial Pacific with that over the east (2.5ºS-2.5ºN, 150ºW-90ºW) equatorial Pacific (magenta boxes in Supplementary Fig. 4), for both observation and model simulations.

### Thermocline-wind coupling

As a key element of the Bjerknes feedback, thermocline-wind coupling determines the strength of thermocline feedback and thus ENSO amplitude[30,50]. Following previous studies[30,50], we estimate the coupling efficiency between the ocean and atmosphere through regression of zonal thermocline slope onto a unit zonal wind anomaly,

$$[H_E] - [H_W] = \beta_H [\tau_x] \tag{1}$$

where $\beta_H$ is the regression coefficient, $[H_E]$ and $[H_W]$ are anomalous thermocline depth averaged over the east (5ºS-5ºN, 140ºW-80ºW) and west (5ºS-5ºN, 120ºE-180º) equatorial Pacific, respectively, and $[\tau_x]$ denotes zonal wind anomalies (TAUU) averaged in the central Pacific (5ºS-5ºN, 170ºE-130ºW). The total thermocline slope response to wind is defined as the response sensitivity ($\beta_H$) multiplied by the standard deviation of $[\tau_x]$. From linear theory of equatorial wave dynamics, a larger $\beta_H$ means a higher energy eastward-propagating Kelvin waves can transmit from wind, resulting in larger swings of thermocline that ultimately promote larger SST anomalies through the thermocline feedback[30,49].

### Statistical significance test

A bootstrap method[58] is used to examine the one-standard-deviation range in associated time evolutions. For each time step, 17 values from the 17 models are resampled randomly to construct 10,000 realizations of the multi-model ensemble mean. In this random resampling process, any model is allowed to be selected again. The standard deviation of the 10,000 inter-realizations of multi-model ensemble mean is used for the uncertainty range (for example, purple shading in Fig. 1a). The bootstrap method is also used to examine whether the difference in the multi-model ensemble mean rainfall change between each century is significant (for example, Fig. 5b), and if the difference of the multi-model mean change between the two periods is greater than the sum of the two separate 10,000-realization standard deviation values, the difference is considered as statistically significant above the 95% confidence level.

## Data availability

Data related to the paper can be downloaded from the following websites. ERSST v5, https://psl.noaa.gov/data/gridded/data.noaa.ersst.v5.html;GPCPv2.3, https://psl.noaa.gov/data/gridded/data.gpcp.html, CMIP6 from https://esgf-node.llnl.gov/search/cmip6/ and https://esgf.ceda.ac.uk/thredds/catalog/esg_cmip6/CMIP6/catalog.html, CMIP5 from https://esgf-node.llnl.gov/search/cmip5/ and https://esgf.ceda.ac.uk/thredds/catalog/esg_dataroot/cmip5/catalog.html.

## Code availability

The codes to calculate results associated with main figures in this study are available upon request.

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

## Acknowledgements
This study is supported by National Key Research & Development Program of China (2023YFF0805200), the Science and Technology Innovation Project of Laoshan Laboratory (LSKJ202203300), National Key Research and Development Program of China (2023YFF0806700) and the Taishan Scholars Program. T.G. is supported by National Natural Science Foundation of China (NSFC) project (42206209) and China National Postdoctral Program for Innovative Talents (BX20220279). F.J. is supported by the National Key Research and Development Program of China (2020YFA0608801), NSFC (42376006), LSKJ202202402, Youth Innovation Promotion Association of Chinese Academy of Sciences (2021205) and tsqn202312265.

## Author contributions
W.C. and T.G. codesigned the study and wrote the initial manuscript in discussion with F.J. and L.W. T.G. performed analysis and generated all figures. All authors contributed to interpreting the results and improving the paper.

## Competing interests
The authors declare no competing interests.
