## [Peer Review File · Nature Communications]

Nonlinear ENSO response to formation of a permanent El Niño-like state under greenhouse warmingREVIEWER COMMENTS

Reviewer #1 (Remarks to the Author):

Review of the manuscript entitled ‘A transition to a permanent El Niño state under persistent global warming’. The manuscript is overall well written. However, there are two important issues. One is related to the argument of the authors that the reduction in ENSO variability is a sign of the persistent El Niño condition. Another point is that the authors need to provide a dynamic analysis to explain the decrease in the intensity of El Niño events in the 22nd and 23rd centuries. Some further comments are listed below. My recommendation is a major revision.

Lines 32-33 and elsewhere: Regarding the development of the persistent El Niño-like warming pattern, as argued by Vecchi and Wittenberg (2010) (<https://doi.org/10.1002/wcc.33>), it is not appropriate to use ‘El Niño-like’ terminology for the greater warming of the eastern tropical Pacific: “However, the usefulness and validity⁵⁰ of the phrase ‘El Niño-like’ may be limited because of the following reasons: the zonal asymmetry in the projected warming across the equatorial Pacific is much smaller than that arising during El Niño,^{43, 51} the mechanisms for these changes are distinct from those of El Niño,⁴⁴⁻⁴⁶ there are many changes in the Pacific that do not resemble those of El Niño,^{46, 50, 51} and—most importantly—there are many climate anomalies over land that do not resemble those during El Niño.^{52, 53} For example, under increased greenhouse gases, the Maritime Continent and Indian Subcontinent are projected to become wetter and southwestern North America drier (Figure 4(b))—all of which are unlike the impacts of El Niño (Figure 1).” So, it may not be possible to conclude that the reduction in ENSO variability is the sign of the persistent El Niño condition.

Line 33-35: Please note that the El Niño-like warming pattern does not necessarily mean the El Niño condition because SST anomalies in the future should be calculated relative to its contemporary climate. So, a permanent El Niño condition cannot be concluded solely based on the greater warming of the eastern tropical Pacific. In addition, Climate models may be seriously deficient in their representation of ENSO or the tropical Pacific because enhanced warming in the eastern Pacific by coupled climate models in the historical period is contrary to what has been observed, which tends to show cooling in the eastern Pacific. There has been a debate in the climate community over this discrepancy for decades, perhaps starting with Cane et al. (1997) (<https://doi.org/10.1126/science.275.5302.957>), and the reasons for the discrepancy are not agreed upon.

Line 58: Please note that the results of some studies do not show any considerable changes in the frequency of strong El Niño events. So, there is not yet consensus in this regard and this needs to be discussed.

Lines 90-91: The Oceanic Niño Index is generally calculated based on centered 30-year base periods and updated every 5 years (e.g. in NOAA Climate Prediction Center: https://origin.cpc.ncep.noaa.gov/products/analysis_monitoring/ensostuff/ONI_v5.php). Any reason that you applied a 51-year base period?

Line 107: Why is the threshold value for the identification of El Niño considered 0.75? The standard threshold value is 0.5 and should be persisted for at least 5 three-month running means (<https://ggweather.com/enso/oni.htm>)

Lines 164-183: The authors need to provide a dynamic analysis to explain the decrease in the intensity of El Niño events in the 22nd and 23rd centuries, which finally led to a decrease in ENSO variability.

Typos:

Line 166: Please replace 'pre-2100' with 'during the pre-2100' and 'century' with 'centuries'

Line 168: Please replace 'The shrinking in non-convective area means that establishment' with 'The shrinking in the non-convective area means that the establishment'

Line 172-175 & 177-181 & 185-188: The sentences are too long, please split them.

Lines 197-198:

Reviewer #2 (Remarks to the Author):

The paper assesses how ENSO is predicted to change in the next 300 years, under high emission scenarios. I have some serious concerns around the data and methods, which might be just due to a lack of information:

I am concerned about the anomalies calculation with 2 running means of apparently arbitrary length (11 and 51 years). There are no references to such a method in the paper, I have never come across it, and the authors do not argue for it. The statement "Using different length of running windows to compute anomaly does not alter our results." does not suffice.

An important piece of information that I cannot find in the manuscript is how the authors have combined the results across the model ensemble to construct the mean they show in most figures. Models are known for having widely different results, particularly under strong emission scenarios and in projections that are further ahead in time (as shown in Extended Data Fig.1), simply averaging across the models might be removing important information on variability.

Which experiments are included in the analysis is also an important piece of information, there should be a table with names, institutions, and pertinent references in the main text so that the reader can access full information. For instance, the experiment from CMIP5 MPI-ESM-LR rcp85 as well as the CMIP6 version, from my search online end with 2100 while in Extended Data Fig.1 there are data-points until 2300, which means the information given in the table is not enough to identify the experiment, and not enough to understand why certain models are included and others are not. Furthermore, I need an explanation as to why CMIP5 and CMIP6 models are combined here. CMIP6 experiments have been made available for some time now, so it cannot be due to a lack of

experiments, nor it is to compare the 2 generations, since the authors do not keep track of the two subsets throughout the analysis.

Furthermore, the structure of the paper needs some re-working. The headings do not help guiding the reader across the analysis, and too many figures are in the extended material, forcing frequent movements "back&forth".

Response to Referee #1

The manuscript is overall well written. However, there are two important issues. One is related to the argument of the authors that the reduction in ENSO variability is a sign of the persistent El Niño condition. Another point is that the authors need to provide a dynamic analysis to explain the decrease in the intensity of El Niño events in the 22nd and 23rd centuries. Some further comments are listed below. My recommendation is a major revision.

Thank you for your positive and helpful comments. Please see our response to your comments below (in colour).

Lines 32-33 and elsewhere: Regarding the development of the persistent El Niño-like warming pattern, as argued by Vecchi and Wittenberg (2010) (<https://doi.org/10.1002/wcc.33>), it is not appropriate to use ‘El Niño-like’ terminology for the greater warming of the eastern tropical Pacific: “However, the usefulness and validity⁵⁰ of the phrase ‘El Niño-like’ may be limited because of the following reasons: the zonal asymmetry in the projected warming across the equatorial Pacific is much smaller than that arising during El Niño,^{43, 51} the mechanisms for these changes are distinct from those of El Niño,⁴⁴⁻⁴⁶ there are many changes in the Pacific that do not resemble those of El Niño,^{46, 50, 51} and—most importantly—there are many climate anomalies over land that do not resemble those during El Niño.^{52, 53} For example, under increased greenhouse gases, the Maritime Continent and Indian Subcontinent are projected to become wetter and southwestern North America drier (Figure 4(b))—all of which are unlike the impacts of El Niño (Figure 1).” So, it may not be possible to conclude that the reduction in ENSO variability is the sign of the persistent El Niño condition.

We agree that some El Niño-like changes are different from those during a typical El Niño event. We borrow this terminology from paleo-climatic studies previously defined to refer to a background warming pattern in which climatological zonal and meridional SST gradients weaken substantially in the equatorial Pacific (e.g., Wara et al., 2005). We have made it clear (lines 69-71) and added a discussion in lines 217-222, citing your quoted study, which read:

“The ‘El Niño-like’ mean condition should not be taken as a state where all climate anomalies resemble those of an El Niño, or that the associated mechanisms for the changes are the same as those of El Niño⁵³. Instead, an “El Niño-like” mean condition here is characterised by a substantially weakened west-minus-east and meridional SST gradients, with only some anomalies resembling those of an El Niño.”

Our inference that “the reduction in ENSO variability is the sign of the persistent El Niño condition” is based on the inter-model relationship in Fig.3c, d, in which models with a greater warming in the eastern equatorial Pacific systematically generate a weaker ENSO variability. The systematic relationship cannot be explained by ENSO rectification onto mean state, because a higher (lower) ENSO variability rectifies on the mean state via nonlinear oceanic temperature advection leading to a larger (smaller) eastern Pacific warming (e.g., Hayashi et al., 2020). Thus, the reduction in ENSO variability is a consequence (not cause) of the El Niño-like warming pattern. As the fast eastern warming proceeds, and shrinking of the non-convective area of the equatorial Pacific continues, establishing convection in the non-

convective area, as during an El Niño, requires smaller convective and SST anomalies. We have added a discussion on this regard. Please see lines 211-217, which read:

“ENSO rectification, in which higher ENSO variability rectifies on the mean state via nonlinear oceanic temperature advection leading to a larger eastern Pacific warming⁵², would not explain the systematic relationship between decreased ENSO variability and enhanced eastern Pacific warming after 2100 (**Fig. 3c and Fig. 5a**). The reduction in ENSO variability is thus a symptom of a developing permanent El Niño-like mean condition characterised by the collapsing west-minus-east and meridional SST gradients (red and blue curve, **Fig. 5a**).”

Line 33-35: Please note that the El Niño-like warming pattern does not necessarily mean the El Niño condition because SST anomalies in the future should be calculated relative to its contemporary climate. So, a permanent El Niño condition cannot be concluded solely based on the greater warming of the eastern tropical Pacific.

We should have highlighted that we calculated SST anomalies relative to its contemporary climatology from 1860-2299 using a 51-year running window; that is, anomalies are referenced to an evolving climatology in the 51-year running window (this information is in the Methods). We have now made it clear (see lines 85-87).

We have changed to a “permanent El Niño-like mean condition”.

In addition, Climate models may be seriously deficient in their representation of ENSO or the tropical Pacific because enhanced warming in the eastern Pacific by coupled climate models in the historical period is contrary to what has been observed, which tends to show cooling in the eastern Pacific. There has been a debate in the climate community over this discrepancy for decades, perhaps starting with Cane et al. (1997) (<https://doi.org/10.1126/science.275.5302.957>), and the reasons for the discrepancy are not agreed upon.

We have added a discussion about the model-observation discrepancy on mean SST warming pattern. Please see lines 125-130 in the revised manuscript, which read:

“There has been a debate as to how the background mean SST of the tropical Pacific may respond to greenhouse warming^{43,44}. Though different from the recently observed^{43,44,45}, evidence suggests that the enhanced eastern Pacific warming pattern in the future is possible as greenhouse effects progressively overwhelm other factors, such as decadal climate variability, which in observations could temporarily mask the warming signal^{46,47}.”

Line 58: Please note that the results of some studies do not show any considerable changes in the frequency of strong El Niño events. So, there is not yet consensus in this regard and this

needs to be discussed.

Internal variability could mask the climate change signal. We have revised the text. Please see lines 50-53.

Lines 90-91: The Oceanic Niño Index is generally calculated based on centered 30-year base periods and updated every 5 years (e.g. in NOAA Climate Prediction Center: https://origin.cpc.ncep.noaa.gov/products/analysis_monitoring/ensostuff/ONI_v5.php). Any reason that you applied a 51-year base period?

The choice of using 51-year base period to calculate anomaly is to accord with the 51-year running window used to examine ENSO variability and associated mean-state changes. We have made it clear in the revised Methods (please see lines 509-510).

Further, we have verified that either using NOAA's Oceanic Niño Index (i.e., centered 30-year base periods and updated every 5 years) or a polynomial fit to detrend anomalies for the whole period (i.e., no pre-assumed time window), does not qualitatively alter our results (**Fig.R1-1**). In addition, our conclusions are not dependent on the time length (11yr, 31yr and 71yr) of running windows to calculate anomalies (**Fig.R1-2**).

We have discussed sensitivity of our results to anomaly definitions (see lines 92-94 and lines 518-522), and included Fig.R1-1 and Fig.R1-2 as new Supplementary Figs.10 and 11 in the revised manuscript.

Fig. R1-1 | Sensitivity of ENSO variability change to anomaly definition. **a, b,** As in Fig. 1c, d, respectively, but for **(a)** 51-year running standard deviation of Niño3.4 SST anomaly (black) and **(b)** 51-year running mean amplitude of Niño3.4 SST anomaly for strong (red) and all (blue) El Niño events using NOAA’s ONI method. Specifically, SST anomalies are calculated based on centered 30-year base periods and updated every 5 years, and an El Niño event is defined as when the three-month running mean ONDJF Niño3.4 index exceeds a value of 0.5 standard deviation calculated from the corresponding running periods. **c, d,** As in Fig. 1c, d, respectively, but using a fifth-order polynomial fit to detrend SST anomalies. The nonlinear ENSO response, with a mild pre-2100 increase followed by a substantial post-2100 decrease, is insensitive to anomaly definitions.

Fig. R1-2 | Sensitivity of ENSO variability change to length of running window. a, b, As in Fig.1c, d, respectively, but for (a, b) 11-year, (c, d) 31-year and (e, f) 71-year running windowed results, in which SST anomaly is constructed with reference to a corresponding length of running windowed DJF climatology. The nonlinear ENSO response, with a mild pre-2100 increase followed by a substantial post-2100 decrease, is insensitive to length of running windows.

Line 107: Why is the threshold value for the identification of El Niño considered 0.75? The standard threshold value is 0.5 and should be persisted for at least 5 three-month running means (<https://ggweather.com/enso/oni.htm>)

We chose this threshold (0.75) after previous studies (Okumura, 2019; see also Fig 3.36 in IPCC report, <https://www.ipcc.ch/report/ar6/wg1/figures/chapter-3/figure-3-36>). Our results still hold using the NOAA's method (i.e., 0.5 threshold for at least 5 three-month running

means). Please see **Fig. R1-1** and our response above. We have added related reference in the revision (line 107).

Lines 164-183: The authors need to provide a dynamic analysis to explain the decrease in the intensity of El Niño events in the 22nd and 23rd centuries, which finally led to a decrease in ENSO variability.

We have calculated the dynamical coupling between thermocline and wind, measured by regression of zonal thermocline slope onto wind anomaly (Kim et al. 2013; Kim et al. 2014). As shown in **Fig. R1-3a**, the thermocline-wind coupling strength undergoes a similar pre-2100 increase followed by a substantial post-2100 reduction. The post-2100 reduction is statistically significant above the 95% confidence level (**Fig. R1-3b**), according to a Bootstrap method. Models that simulate a smaller wind anomaly during the 22nd and 23rd centuries systematically generate a weaker thermocline total response to wind (**Fig. R1-3c**), which eventually leads to weaker El Niño SST variability through the thermocline feedback (Kim et al. 2013; Kim et al. 2014; **Fig. R1-3d**). These results indicate that reduced wind variability from the decreased potential intensity and non-convective area leads to reduced post-2100 ENSO variability through weakened thermocline response to the wind, feeding into dynamical ocean-atmosphere coupling.

In the revised version, we have added a new section ‘Thermocline-wind coupling’ in Methods and associated analysis in the main text (lines 199-206). The original Fig.4 has been split into a new Fig.4 combining Fig.R1-3, and a new Fig.5.

Fig. R1-3 | Reduced thermocline-wind coupling curtails post-2100 ENSO variability. **a**, 51-year running mean response sensitivity coefficient from regression of normalised zonal thermocline slope (east minus west) anomalies onto normalised TAUU anomalies, from 1860 to 2300 under a high-emission scenarios. Years on the x-axis denote the end year of the running window. **b**, Multi-model histograms of 10, 000 realizations of a bootstrap method for the thermocline slope response sensitivity to wind in 1860-1909 (blue bars) and 2250-2299 (red bars). Solid lines and shadings in **a**, **b** indicate multi-model mean and 1.0 standard deviation of a total of 10,000 inter-realizations based on a Bootstrap method, respectively. **c**, Inter-model relationship between TAUU variability and total response of thermocline slope to wind, calculated as the thermocline response sensitivity coefficient multiplied by the standard deviation of TAUU. **d**, Inter-model relationship between total response of thermocline slope to wind and normalized Niño3.4 SST standard deviation. All the indices are calculated over the DJF season and normalized by its whole 400-year (1900-2299) standard deviation (s.d.). Linear fit (solid black line) is displayed together with correlation coefficient r and corresponding p value. Red star denotes CESM2-WACCM.

Typos:

Line 166: Please replace 'pre-2100' with 'during the pre-2100' and 'century' with 'centuries'

Line 168: Please replace ‘The shrinking in non-convective area means that establishment’ with ‘The shrinking in the non-convective area means that the establishment’

Line 172-175 & 177-181 & 185-188: The sentences are too long, please split them.

All done.

Thank you again for your helpful and thoughtful comments.

References:

Wara, M. W., Ravelo, A. C., & Delaney, M. L. Permanent El Niño-like conditions during the Pliocene warm period. *Science*, **309**, 758-761 (2005).

Hayashi, M., Jin, F.-F. & Stuecker, M. F. Dynamics for El Niño-La Niña asymmetry constrain equatorial-Pacific warming pattern. *Nat. Commun.* **11**, 4230 (2020).

Okumura, Y. M. ENSO diversity from an atmospheric perspective. *Curr Clim Change Rep* **5**, 245-257 (2019).

Kim, S. T., Cai, W., Jin, F.-F. & Yu, J.-Y. ENSO stability in coupled climate models and its association with mean state. *Clim. Dyn.* **42**, 3313–3321 (2013).

Kim, S. T. et al. Response of El Niño sea surface temperature variability to greenhouse warming. *Nat. Clim. Chang.* **4**, 786–790 (2014).

Response Referee #2

The paper assesses how ENSO is predicted to change in the next 300 years, under high emission scenarios. I have some serious concerns around the data and methods, which might be just due to a lack of information:

Thank you for your positive and helpful comments. Please see our response below (in blue).

I am concerned about the anomalies calculation with 2 running means of apparently arbitrary length (11 and 51 years). There are no references to such a method in the paper, I have never come across it, and the authors do not argue for it. The statement "Using different length of running windows to compute anomaly does not alter our results." does not suffice.

The choice of using 51-year base period to calculate anomaly is to accord with the 51-year running window used to examine ENSO variability and associated mean state changes. Subtracting an 11-year running mean is to eliminate any influence from variability on decadal and longer time scales, therefore the resultant anomaly time series only contains interannual variability. We have added an explanation on these choices in the revision (please see lines 509-513).

We have also verified that using NOAA's Oceanic Niño Index (https://origin.cpc.ncep.noaa.gov/products/analysis_monitoring/ensostuff/ONI_v5.php) or a polynomial fit to detrend anomalies for the whole period (i.e., no pre-assumed time window), does not qualitatively alter our results (**Fig.R2-1**). In particular, observed SST anomalies calculated using our method and that of NOAA's match well with each other, with a strong correlation ($r=0.99$) in 1900-2023. Further, our conclusions are insensitive to the time length (11yr, 31yr and 71yr) of running windows for calculating anomalies (**Fig.R2-2**).

We have discussed sensitivity of our results to anomaly definitions (see lines 91-93 and lines 514-518) and included Fig.R2-1 and Fig.R2-2 as new Supplementary Fig.10 and 11 in the revised manuscript.

Fig. R2-1 | Sensitivity of ENSO variability change to anomaly definition. **a, b,** As in Fig. 1c, d, respectively, but for **(a)** 51-year running standard deviation of Niño3.4 SST anomaly (black) and **(b)** 51-year running mean amplitude of Niño3.4 SST anomaly for strong (red) and all (blue) El Niño events using NOAA’s ONI method. Specifically, SST anomalies are calculated based on centered 30-year base periods and updated every 5 years, and an El Niño event is defined as when the three-month running mean ONDJF Niño3.4 index exceeds a value of 0.5 standard deviation calculated from the corresponding running periods. **c, d,** As in Fig. 1c, d, respectively, but using a fifth-order polynomial fit to detrend SST anomalies. The nonlinear ENSO response, with a mild pre-2100 increase followed by a substantial post-2100 decrease, is insensitive to anomaly definitions.

Fig. R2-2 | Sensitivity of ENSO variability change to length of running window. **a, b,** As in Fig.1c, d, respectively, but for **(a, b)** 11-year, **(c, d)** 31-year and **(e, f)** 71-year running windowed results, in which SST anomaly is constructed with reference to a corresponding length of running windowed DJF climatology. The nonlinear ENSO response, with a mild pre-2100 increase followed by a substantial post-2100 decrease, is insensitive to length of running windows.

An important piece of information that I cannot find in the manuscript is how the authors have combined the results across the model ensemble to construct the mean they show in most figures. Models are known for having widely different results, particularly under strong emission scenarios and in projections that are further ahead in time (as shown in Extended Data Fig.1), simply averaging across the models might be removing important information on variability.

We should have made it clear. Evolution of ENSO variability and mean state changes are calculated first in each model in a 51-year running window that moves from 1860 to 2299 and then averaged across the models.

An arithmetic multi-model average is a commonly used approach in the community to eliminate the influence of internal variability that is random and vastly different across the models.

We have added associated information in various parts of the revision (lines 90-92 and lines 140-141).

Which experiments are included in the analysis is also an important piece of information, there should be a table with names, institutions, and pertinent references in the main text so that the reader can access full information. For instance, the experiment from CMIP5 MPI-ESM-LR rcp85 as well as the CMIP6 version, from my search online end with 2100 while in Extended Data Fig.1 there are data-points until 2300, which means the information given in the table is not enough to identify the experiment, and not enough to understand why certain models are included and others are not. Furthermore, I need an explanation as to why CMIP5 and CMIP6 models are combined here. CMIP6 experiments have been made available for some time now, so it cannot be due to a lack of experiments, nor it is to compare the 2 generations, since the authors do not keep track of the two subsets throughout the analysis.

As stated in the Methods (line 496), we used all the 17 (9 CMIP5 and 8 CMIP6) models that are available so far with outputs to 2300 under RCP85/SSP585.

Information of the MPI-ESM-LR model that ends at 2300 can be found from: https://esgf.ceda.ac.uk/thredds/catalog/esg_dataroot/cmip5/output1/MPI-M/MPI-ESM-LR/rcp85/mon/ocean/Omon/r1i1p1/latest/tos/catalog.html

We view CMIP5 and CMIP6 as a whole and combine them together, because of the limited number of models in each generation with outputs to 2300 (though there are many with outputs to 2100), and because of the similarity between RCP and SSP emission pathways. Individual model results are given in Supplementary Fig.1 and Fig. 6.

We have revised the table following your suggestion. Given the length of the table, we feel it appropriate to put the table in the supplementary.

Furthermore, the structure of the paper needs some re-working. The headings do not help guiding the reader across the analysis, and too many figures are in the extended material, forcing frequent movements "back&forth".

We have revised various parts of the text to make it easy to follow. Thank you again for your helpful and thorough comments.

REVIEWERS' COMMENTS

Reviewer #1 (Remarks to the Author):

The authors properly addressed my comments and I recommend the acceptance of the manuscript.

Reviewer #2 (Remarks to the Author):

Dear Authors,

I appreciate how you have taken into consideration all of my comments and addressed the concerns I raised. I have 2 short comments/questions left:

I still believe that the decision for a 51 year long mean is not entirely justified or referenced. you state "The choice of using 51-year base period to calculate anomaly is to accord with the 51-year running window used to examine ENSO variability and associated mean state changes." is kind of a circular argument.

I wonder about the structure of the article where the Methods are at the very end of the paper, after the References. Why is that?

Response to Referee #1

The authors properly addressed my comments and I recommend the acceptance of the manuscript.

Thank you!

Response Referee #2

I appreciate how you have taken into consideration all of my comments and addressed the concerns I raised.

Thank you for your positive and helpful comments. Please see our response below (in blue).

I have 2 short comments/questions left:

I still believe that the decision for a 51 year long mean is not entirely justified or referenced. you state "The choice of using 51-year base period to calculate anomaly is to accord with the 51-year running window used to examine ENSO variability and associated mean state changes." is kind of a circular argument.

A 51-year long mean is used for the purpose of illustration. We have verified that our conclusions are insensitive to the time length (11yr, 31yr and 71yr) of running windows.

Please see various parts of the revised manuscript (lines 92-94 and lines 293-297) and Supplementary Fig.10 and 11.

I wonder about the structure of the article where the Methods are at the very end of the paper, after the References. Why is that?

The Methods section has been moved ahead of the References section.